# Do beta-amyloid-targeted interventions improve cognition, physical functioning, and overt behaviour of Alzheimer's Disease (AD) patients: Protocol for meta-analysis of Phase 3 clinical trials both completed and terminated

**Chelsea Ann Stellick (Bedrejo)**[1]*, **Andrew Greenshaw**[1], **Mike Paulden**[2], **Sidney Yap**[1], **R. Tyler Marshall**[3], **Janice Y. Kung**[4], **Eldon Spackman**[5]

1 Department of Psychiatry, Faculty of Medicine and Dentistry, University of Alberta, Edmonton, Alberta, Canada, 2 School of Public Health, University of Alberta, Edmonton, Alberta, Canada, 3 Department of Computer Science, Faculty of Science, University of Alberta, Edmonton, Alberta, Canada, 4 John W. Scott Health Sciences Library, University of Alberta, Edmonton, Alberta, Canada, 5 Department of Community Health Sciences, Cumming School of Medicine, University of Calgary, Calgary, Alberta, Canada

* bedrejo@ualberta.ca

**Data Availability Statement:** No datasets were generated or analysed during the current study. All

## Abstract

Accumulation of amyloid-beta (Aβ) in the brain has been explored as a primary cause of Alzheimer's Disease (AD). Better known as the amyloid hypothesis, it has been the main target of researchers vying to bring their therapeutic interventions to market despite several failed attempts by predecessors. In June 2021, *Aduhelm* (Aducanumab) became the first U.S. Food and Drug Administration (FDA) approved treatment for AD based on the amyloid hypothesis in which sparked controversy. This meta-analysis aims to investigate the efficacy of amyloid-beta targeting interventions at all stages of the disease including the prodromal or mild cognitive impairment (MCI) stage compared to placebo. All completed and terminated Phase III trials are assessed to provide a comprehensive overview of interventions targeting amyloid-beta to inform the legitimacy of the amyloid hypothesis.

## Introduction

Alzheimer's Disease (AD) is a progressive neurodegenerative disease that slowly erodes a person's cognitive function, ability to carry out daily tasks and overall behaviour. In 2021, it was estimated 6.5 million Americans aged 65 plus live with the disease with a significant projected increase year over year [1]. No effective treatment yet exists targeting the pathology of AD, beyond providing temporary symptomatic relief and cognitive stabilization with acetylcholinesterase inhibitor (AChEI) Drugs [2]. However, while such temporary relief can improve the quality of life (QoL) of patients, it does not necessarily result in the complete restoration and improvement of one's cognitive, behavioural or physical functioning [2, 3].

relevant data from this study will be made available upon study completion.

**Funding:** The author(s) received no specific funding for this work.

**Competing interests:** The authors have declared that no competing interests exist.

Over the past two decades, researchers have been pushing to discover and commercialize pharmacological interventions to modify the progression of the disease–most notably targeting the abnormal accumulation of amyloid-beta (Aβ) in the brain. The pathophysiologic changes underlying diagnosis and distinguishing Alzheimer's disease from other non-Alzheimer causes of cognitive impairment are 1) the extracellular deposition of the Aβ peptide (A), 2) the intracellular neurofibrillary tau tangles (T) and 3) neurodegeneration (N). Developed by the National Institute on Aging and the Alzheimer's Association, the A/T/N classification provides a framework for drug development and clinical trials highlighting Aβ deposition as an important component of the pathology of Alzheimer's [4]. Aβ deposits and neurofibrillary tau tangles are the two hallmark pathologies of AD diagnosis. However, historically, there has been an emphasis on Aβ deposits in all stages of the disease including the preclinical stage of the disease.

The deposition of Aβ which is measured either by amyloid positron emission tomography (PET) imaging of amyloid plaques or by the concentration of beta-amyloid-42 (Aβ42) in the cerebrospinal fluid (CSF-Aβ42), has been the major focus of drug developers over the last decade and a half. First introduced by George Glenner in 1984, the amyloid hypothesis was later refined by neurogeneticist, John Hardy, who discovered that mutations in the amyloid precursor protein (APP) ultimately lead to the abnormal accumulation of amyloid-beta in the brain [5, 6]. Since this discovery, the amyloid hypothesis has been the major target in the pharmaceutical industry, over other theories such as the tau hypothesis and the cholinergic hypothesis.

Treatment candidates that target amyloid beta include secretase inhibitors and amyloid beta immunotherapies (passive and active). Secretase inhibitors, such as beta-secretase 1 (BACE1) inhibitors and gamma-secretase inhibitors (GSIs) will eliminate the production of amyloid beta. Immunotherapy, on the other hand, uses the body's own immune system to help clear the amyloid-beta peptides that lead to plaques. Despite the failure to advance toward commercialization for several anti-amyloid candidates, the amyloid hypothesis remains the prominent hypothesis for AD pathology.

Currently, the amyloid hypothesis remains contentious, especially with the approvals by the United States Food and Drug Administration (FDA) of *Aduhelm* (Aducanumab), *Leqembi* (Lecanemab) and most recently, *Kisunla* (Donanemab). Among the two-Phase III studies for Aduhelm in patients with mild cognitive impairment (MCI) and early-stage dementia, ENGAGE and EMERGE, only EMERGE demonstrated that it had met its primary endpoint (change in Clinical Dementia Rating (CDR) Sum of Boxes score), at higher dosages in a post-hoc analysis. No drug benefi was demonstrated in the ENGAGE trial. While both trials demonstrated a significant reduction in Aβ in the trials, its impact on cognition remained in question. Some subject matter experts believe the FDA's approval of Aduhelm was the first key step forward in AD research, while others disagree. To further put the amyloid hypothesis under a critical lens, a body of influential work exploring a form of amyloid called Aβ*56 fell under scrutiny in July 2022, for potentially fraudulent activity relating to the manipulation of images reported in the peer-reviewed scientific literature [7]. Despite the contention, the FDA approval of Leqembi followed in January of 2023. That said, Leqembi also sparked some controversy and uncertainty in regard to its efficacy. Some experts agree that its efficacy profile is modest especially in comparison to its safety concerns and high price tag. Eli Lilly and Co.'s monoclonal antibody, Donanemab has recently been approved and has demonstrated some promise for anti-amyloid therapeutics.

Current controversies surround Aduhelm and accusations around potential fraudulent images found in a series of articles that influenced many researchers towards pushing the momentum of the amyloid-oligomer hypothesis. This context in addition to the recent developments around Eli Lilly's Donanemab, makes it essential to review the evidence from all trials including the terminated and ongoing studies before giving up hope for efficacy of beta-

amyloid interventions. This meta-analysis will provide a comprehensive assessment of all completed and terminated Phase III pharmacological interventions targeting beta-amyloid for AD to provide further insights and learnings to help better inform the legitimacy of the amyloid hypothesis and whether further research should continue to be conducted or if efforts should be deployed towards a different hypothesis.

## Methods and analysis

### Study guidelines and registration

This protocol will be conducted in accordance with the Preferred Reporting Items for Systematic Review and Meta-Analysis protocol (PRISMA-P) statement [8]. The systematic review and meta-analysis will follow the guidelines set out in the Cochrane Handbook [9]. The study will be conducted from February 1, 2024, to November 1, 2024 and reported following the PRISMA statement [10]. This protocol is registered in the International Prospective Register of Systematic Reviews (CRD42022382948).

## Information sources

### Search strategy

The search strategy was designed and conducted in consultation with an experienced medical librarian (JYK) who conducted comprehensive searches in MEDLINE, EMBASE, PsycINFO, Cochrane Library (including the Cochrane Register of Controlled Trials and the Cochrane Database of Systematic Reviews), and Scopus. Additional studies will be searched through ClinicalTrials.gov to incorporate ongoing, completed, or unpublished/terminated trials. We will search through the AlzForum website (https://www.alzforum.org) for comprehensive summaries of each therapeutic. Finally, the reference lists of each publication that enters full-text screening will be reviewed. No language or date limits were applied. The search strategy for MEDLINE is presented in Table 1.

### Eligibility criteria

Studies will be included in the systematic review and meta-analysis as outlined by the following PICOs criteria outlined below and on Table 2.

### Types of participants

Patients in the trial should be diagnosed with Alzheimer's Disease as outlined by reported and well-defined diagnostic criteria, including the National Institute of Neurological and Communicative Disorders and Stroke/Alzheimer's Disease and Related Disorders Association (NINCDS-ADRDA criteria) [11] and 2011 National Institute on Aging/Alzheimer's Association (NIA-AA Criteria) [12]. All stages of AD will be included in the review if clear descriptions are provided. There will be no restrictions on demographic factors such as age, sex, education level, combined medication and so on.

### Types of interventions

The intervention in the treatment group should be any anti-Aβ drug that either decreases Aβ production such as secretase inhibitors, prevents Aβ aggregation, such as amyloid beta aggregation inhibitors, and finally increases Aβ removal, such as amyloid beta immunotherapies (passive and active). This includes intravenous immunoglobulin. Natural agents such as dietary

**Table 1. Search strategy for Ovid MEDLINE® ALL 1946 to September 30, 2022.**

| | |
|---|---|
| 1. | alzheimer*.mp. |
| 2. | exp Alzheimer Disease/ |
| 3. | exp Dementia/ |
| 4. | dementia*.ti,ab,kf. |
| 5. | 1 or 2 or 3 or 4 |
| 6. | (amyloid beta or beta amyloid).mp. |
| 7. | (amyloidbeta or betaamyloid).mp. |
| 8. | exp Amyloid beta-Peptides/ |
| 9. | amyloid hypothes*.mp. |
| 10. | (amyloid plaque* or senile plaque*).mp. |
| 11. | exp Plaque, Amyloid/ |
| 12. | amyloid deposit*.mp. |
| 13. | (anti-amyloid or antiamyloid).mp. |
| 14. | 6 or 7 or 8 or 9 or 10 or 11 or 12 or 13 |
| 15. | 5 and 14 |
| 16. | randomized controlled trial.pt. |
| 17. | clinical trial.pt. |
| 18. | randomi?ed.ti,ab. |
| 19. | placebo.ti,ab. |
| 20. | dt.fs. |
| 21. | randomly.ti,ab. |
| 22. | trial.ti,ab. |
| 23. | groups.ti,ab. |
| 24. | or/16-23 |
| 25. | animals/ |
| 26. | humans/ |
| 27. | 25 not (25 and 26) |
| 28. | 24 not 27 |
| 29. | 15 and 28 |
| 30. | (veterinary or rabbit or rabbits or animal or animals or mouse or mice or rodent or rodents or rat or rats or hamster* or pig or pigs or porcine or horse* or equine or cow or cows or bovine or goat or goats or sheep or ovine or canine or dog or dogs or feline or cat or cats or zebrafish).ti. |
| 31. | 29 not 30 |
| 32. | (phase 3 or phase III or phase three or third phase or 3rd phase).mp. |
| 33. | 31 and 32 |

measures will not be included. Interventions (solely) targeting other pathologies of AD will not be included. We included all routes of administration and dose regimens in our review.

## Types of comparisons

Participants should only be treated with a placebo of any type if it is clearly described. Subjects in the control group must receive the placebo with the same mode of administration and frequency as the experimental group. Participants should have similar baseline characteristics as the intervention group.

## Types of outcome measures

Included studies must report primary outcomes that measure either cognitive, behavioural and/or physical functioning. The Alzheimer's Disease Assessment Scale-Cognitive scale

**Table 2. PICOs criteria.**

| Included | Excluded |
|---|---|
| **Population**<br>• Male or Female (age 65+)<br>• Diagnosis of probable with Alzheimer's Disease as outlined by reported and well-defined diagnostic criteria such as the National Institute of Neurological and Communicative Disorders and Stroke/Alzheimer's Disease and Related Disorders Association (NINCDS-ADRDA criteria).<br>• All stages of AD (mild to severe) including early AD (prodromal, mild cognitive impairment, etc.) | **Population**<br>• Studies where patients are suspected of Alzheimer's Disease<br>• Studies where patients are "at risk" of Alzheimer's Disease |
| **Intervention**<br>• Any anti-Aβ drug that either decreases Aβ production, prevents Aβ aggregation, or increases Aβ removal.<br>• Intravenous immunoglobulin is included | **Intervention**<br>• Studies where Aβ was not the primary target of the intervention.<br>• Natural agents (dietary adjustments, exercise etc.) |
| **Control**<br>• Placebo only | **Control**<br>• Not placebo<br>• Uncontrolled studies |
| **Outcome**<br>• Primary cognitive function: Alzheimer's Disease Assessment Scale-Cognitive scale (ADAS-Cogs), Clinical Dementia Rating Scale-Sum of Boxes (CDR-SB)<br><br>• Secondary cognitive function: Mini-Mental State Examination (MMSE) score.<br>• Primary physical function: Alzheimer's Disease Cooperative Study–Activities of Daily Living Scale (ADCS-ADL) score<br>• Primary behavioral function: Neuropsychiatric Inventory (NPI)<br>• Secondary (exploratory) outcome measures: Quality of Life—Alzheimer's Disease scale (QoL-AD) and the EuroQol-5D (EQ-5D) and severe adverse events (SAE's)<br>• Secondary (exploratory) biomarker outcome measures: mean and/or percent change from baseline in Aß40, Aß42, sAPPß, Total Tau and Phosphorylated tau. | **Outcome**<br>• Studies that did not report a validated outcome measurement scale for cognitive, behavioral, or physical function as a primary outcome.<br>• Studies that only reported changes in biomarkers (Cerebrospinal fluid (CSF) indications of beta-amyloid 42, Tau, and phospho-tau) as a primary outcome |
| **Study Type**<br>• **Phase III randomized controlled trials**. | **Study Type**<br>• Phase I or II randomized controlled trials<br>• Active trials<br>• Uncontrolled trials<br>• Non-randomized trials<br>• Observational studies<br>• Qualitative studies<br>• Case reports<br>• Expert opinion articles<br>• Grey literature<br>• Systematic reviews |

(ADAS-Cogs) [13] and all its versions and Global Impression scores to evaluate the pharmacological effects of the intervention, such as the Clinical Dementia Rating Scale-Sum of Boxes (CDR-SB) [14] will be the primary cognitive outcomes of interest. Other scales measuring cognitive function such as the Mini-Mental State Examination (MMSE) score [15] will be recorded as secondary outcomes. Our primary outcome to measure competence in basic daily

activities (physical function) is the Alzheimer's Disease Cooperative Study–Activities of Daily Living Scale (ADCS-ADL) score [16]. Finally, the primary outcome for the assessment of neuropsychiatric symptoms (behavioural functioning) is the Neuropsychiatric Inventory (NPI) [17]. Secondary outcomes of interest include quality of life (QoL) measured by the Quality of Life—Alzheimer's Disease scale (QoL-AD) [18, 19], and the EuroQol-5D (EQ-5D) [20], and safety indexes such as severe adverse events (SAE's), amyloid-related imaging abnormalities (ARIA), and death. We are also interested in changes in biomarkers (Cerebrospinal fluid (CSF) indications of beta-amyloid 42, beta-amyloid 40, soluble beta-amyloid precursor protein (sAPPß) total tau, and phospho-tau) therefore, changes in biomarkers will be included in our study as exploratory endpoints.

## Types of studies

Only randomized controlled studies (cluster and parallel design) that have been completed or terminated and in Phase III or II/III will be included. Active, recruiting, and ongoing trials without reported results are excluded. Cross-over trials, case series, case controls, one-arm trials, non-randomized trials, cross-sectional, cohort studies or case reports are not included. Trials in preclinical, Phase I or Phase II are not included. Non-primary research (reviews, commentaries, poster presentations, letters, editorials, etc.) will not be included. Extension studies will be excluded.

## Study selection

To facilitate the screening process, the team will use Covidence (see www.covidence.org), a web-based tool designed to support systematic review screening and manage references and full-text PDFs. Duplicates will be removed by the Covidence software. Two independent reviewers (CS and SY) will screen the titles and abstracts extracted from each electron database on Covidence. Literature that did not match the eligibility criteria will be excluded from the study and recorded accordingly. CS and SY will then review the title and abstracts and excluded articles that did not match the inclusion/exclusion criteria. A third reviewer (TM) will be consulted when a consensus is not reached between the two reviewers. Any discrepancies will be recorded by the reviewers. The selection procedure is outlined in Fig 1 PRIMSA flow diagram.

## Data extraction

Using a predetermined template, data from each of the included articles will be independently extracted by both reviewers (CS and SY). Included articles will be matched with their appropriate trial from clinicaltrials.gov. If a trial was included from clinicaltrials.gov without an accompanying journal article, we will still include it in our analysis. Similarly, if an article was included in the screening but its trial registered on clinicaltrials.gov was not picked up, we will include both in our analysis. We will include information on Trial name, disease severity, drug name, administration, frequency, age, percentage male, percentage APOE4, and primary outcomes and secondary outcomes. Results will be extracted as least squares mean (standard error), p values and sample sizes for continuous variables. In the event of missing data, corresponding authors were contacted. Any discrepancies between the two reviewers were resolved by consensus and by consulting a third reviewer (TM). Table 3 outlines our data and information extraction schedule.

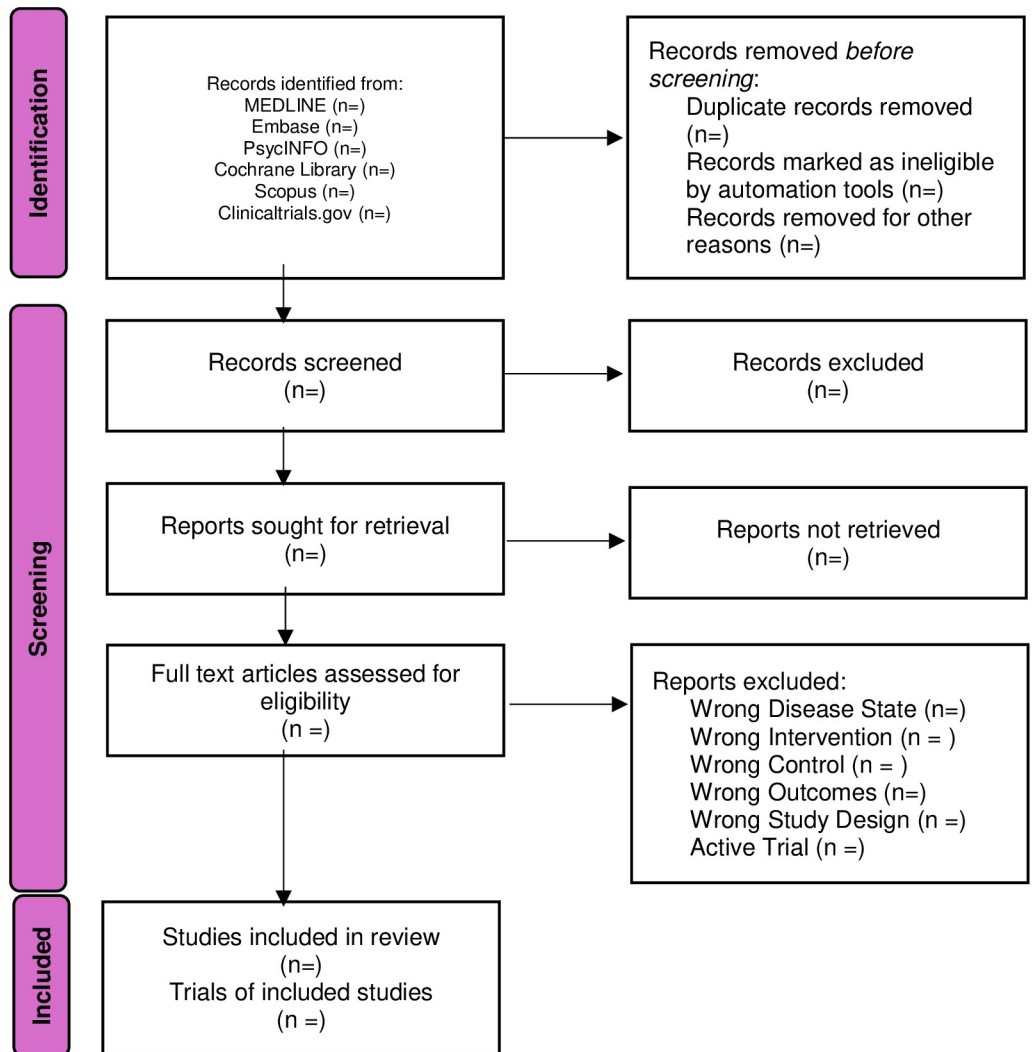

**Fig 1. PRISMA flow chart.** The Preferred Reporting Items for Systematic reviews and Meta-Analyses (PRISMA) flow chart will be used once the results are identified from the selected databases. A record of the screened, excluded and included studies will also be presented in the PRISMA flow chart.

## Data synthesis

### Qualitative synthesis

Data will be collected and synthesized in a table. Specifically, information pertaining to the author, corresponding clinical trial, corporate sponsorship, drug name, drug developer, drug type, sample size and outcome measurements for each individual study.

### Quantitative synthesis

We will conduct a quantitative analysis for each outcome which will be synthesized using RevMan version 5. For continuous variables, we will calculate a mean difference or standard mean difference with a 95% confidence interval. For dichotomous variables, we calculate the risk ratio with a 95% confidence interval. We will calculate the $I^2$ to test for heterogeneity for each pooled result. We will pool our results based on each outcome of interest (ADAS-Cog 11,

**Table 3. Data and information extraction schedule.**

| Subject | Content |
|---------|---------|
| **Publication** | First Author (last name), publish year, corresponding author, contact email, country, corporate sponsorship, percentage of authors from sponsoring company, corresponding clinical trial number, clinical trial name |
| **Participant** | Recruitment source, sample size, age, sex, disease severity, diagnostic criteria, inclusion and exclusion criteria |
| **Intervention** | Drug name, drug developer, administration, dosage and frequency, type of amyloid-beta intervention, patient count assigned to intervention at each dosage, given names of intervention |
| **Control** | Placebo, administration, dosage and frequency, patient count assigned to placebo |
| **Outcome** | Primary and secondary outcome measurements and each time points, primary and secondary outcome measurements at each dosage level, percentage of severe adverse events, percentage of ARIA-E, percentage of ARIA-H, and detailed data |
| **Study Design** | Study start date, study end date, study duration, study termination date (if applicable), study sites, application of randomization and blinding, descriptions about statistical analysis, sample size calculation |
| **Other information** | Dropout rate, reason for withdrawal, reason for trial termination (if applicable), authors' conclusions, limitations |

ADAS-Cog 12, ADAS-Cog 13, ADAS-Cog 14, CDR-SB, MMSE, EQ-5D, EQ-5D (Proxy), QoL-AD, ADCS-ADL, and NPI, NPI-Q, NPI-NH). Using a random-effects model we can look at the average treatment effects for all comparisons. We will present our results using a forest plot and tables.

## Risk of bias

The Cochrane Risk of Bias Tool [21] will be used by the two reviewers (CB and SY) to independently assess the risk of bias for each study. The reviewers will assess each study based on random sequence generation, allocation concealment, blinding of participants, blinding of outcome assessment, incomplete outcome data, selective outcomes reporting and any other sources of bias. The Revised Cochrane risk-of-bias tool for randomized trials (RoB 2) short version (cribsheet) [22], 2019 version will be used to guide the ROB assessment for each included study. Disagreements will be resolved by discussion. The results of the risk-of-bias assessment will be produced in Microsoft Excel using the 22 August 2019 version of RoB2 for randomized controlled trials and later uploaded to robvis, a data visualization tool for risk-of-bias assessments. Using the Grading of Recommendations Assessment, Development, and Evaluation (GRADE) tool [23] the quality of the outcome evidence was assessed. Disagreements will be resolved by discussion and a third, independent reviewer will be available to settle any disagreements.

## Subgroup analysis

If significant heterogeneity is observed, we will perform a subgroup analysis based on disease severity and drug type.

## Discussion

To our knowledge, this is the first systematic review and meta-analysis that quantitatively and qualitatively analyzes Phase III clinical trials for anti-Aβ interventions assessing not only the cognitive domains but also the physical and behavioural domains. It is also the only systematic review to our knowledge that includes all disease stages including mild cognitive impairment and preclinical AD. Finally, it is the only systematic review that we know of that includes small

molecular compounds that reduce Aβ production. We believe the results of our analysis will be further used to collect data, specifically the hazard ratios from outcome results that will be used in an economic model to evaluate the cost-effectiveness of Aβ therapies for both successful and failed Phase III clinical trials. While we supplement our review with some grey literature from AlzForum, we will rely on our searches from databases and ClinicalTrials.gov for our review, which may be limiting as important studies could be missed.

## Supporting information

**S1 Checklist. Preferred reporting items for systematic review and meta-analysis protocols 2015 checklist: Recommended items to address in a systematic review protocol.** (DOCX)

## Author Contributions

**Data curation:** Janice Y. Kung.

**Formal analysis:** Sidney Yap.

**Investigation:** Sidney Yap.

**Methodology:** Janice Y. Kung.

**Supervision:** Andrew Greenshaw, Mike Paulden, Eldon Spackman.

**Writing – original draft:** Chelsea Ann Stellick (Bedrejo).

**Writing – review & editing:** Chelsea Ann Stellick (Bedrejo), Sidney Yap, R. Tyler Marshall.

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
