## [Decision Letter · Decision Letter 0]

22 May 2023

PONE-D-23-05935Do pharmacological interventions targeting beta-amyloid improve cognition, physical functioning, and overt behaviour of Alzheimer’s Disease (AD) patients: A protocol for a meta-analysis of Phase 3 clinical trials both completed and terminatedPLOS ONE

Dear Dr. Stellick (Bedrejo),

Thank you for submitting your manuscript to PLOS ONE. After careful consideration, we feel that it has merit but does not fully meet PLOS ONE’s publication criteria as it currently stands. Therefore, we invite you to submit a revised version of the manuscript that addresses the points raised during the review process. Please submit your revised manuscript by Jul 06 2023 11:59PM. If you will need more time than this to complete your revisions, please reply to this message or contact the journal office at plosone@plos.org. Please include the following items when submitting your revised manuscript:A rebuttal letter that responds to each point raised by the academic editor and reviewer(s). You should upload this letter as a separate file labeled 'Response to Reviewers'.A marked-up copy of your manuscript that highlights changes made to the original version. You should upload this as a separate file labeled 'Revised Manuscript with Track Changes'.An unmarked version of your revised paper without tracked changes. You should upload this as a separate file labeled 'Manuscript'.

We look forward to receiving your revised manuscript.

Kind regards,

Kensaku Kasuga

Academic Editor

PLOS ONE

2. Please ensure all language is factually accurate. Specifically, as many of the contentious amyloid papers are under active investigation, "fraud" cannot be definitively stated. Therefore, please modify the language used on page 4 (e.g., to "potentially fraudulent").

Additional Editor Comments:

As a reviewer points out, the preclinical stage is not “based on A+T-N- classification, indicating the presence of Aβ deposits without the presence of tau tangles and neurodegeneration” as author described. Please carefully review the definition of preclinical AD.

As another reviewer pointed out, biomarker information is essential. Therefore, authors should include these biomarkers to outcome measures in meta-analysis.

Reviewers' comments:

Reviewer's Responses to Questions

**Comments to the Author**

1. Does the manuscript provide a valid rationale for the proposed study, with clearly identified and justified research questions?

Reviewer #1: Yes

Reviewer #2: Yes

2. Is the protocol technically sound and planned in a manner that will lead to a meaningful outcome and allow testing the stated hypotheses?

Reviewer #1: Partly

Reviewer #2: Yes

3. Is the methodology feasible and described in sufficient detail to allow the work to be replicable?

Reviewer #1: Yes

Reviewer #2: Yes

4. Have the authors described where all data underlying the findings will be made available when the study is complete?

Reviewer #1: Yes

Reviewer #2: Yes

5. Is the manuscript presented in an intelligible fashion and written in standard English?

Reviewer #1: Yes

Reviewer #2: Yes

6. Review Comments to the Author

You may also provide optional suggestions and comments to authors that they might find helpful in planning their study.

Reviewer #1: This manuscript addresses a protocol for a forthcoming meta-analysis of anti-Aβ therapy for Alzheimer's disease. While the results of the meta-analysis are important, it is difficult to judge the significance of publishing the protocol because the method itself of the meta-analysis is not novel and only items are modified.

Although grouped together as anti-Aβ drugs in the protocol, the mechanisms of drugs that inhibit Aβ production and those that promote Aβ clearance are very different, and I am concerned that the scientific significance of the results may be diminished by grouping them together as "anti-Aβ drugs".

Overall, it seems that the authors' understanding of Alzheimer's disease and its drug development is somewhat lacking, and the introduction section contains incorrect information and misleading representations.

- The preclinical stage is just the clinically asymptomatic stage, which can include A+T+N- and even A+T+N+ as well as A+T-N-.

- "BETA1" is a typo for BACE1 (beta-site amyloid precursor protein cleaving enzyme 1).

- Although many have criticized the approval of Aduhelm because of the ambiguity of the decision criteria by the FDA, I guess no one has described it as "it is a step backwards".

- At the time of submission, Lecanemab was already approved in addition to Aducanumab, so that should also be listed.

methods section:

- Since phase III trials of anti-Aβ drugs were not initiated until 2007 or later, it is more likely that the 2011 NIA-AA criteria are used in the trials rather than the older NINCDS-ADRDA criteria.

- CDR-SB is a composite of cognitive function and global function (probably equal to "physical function" in the article), so for those familiar with clinical trials in Alzheimer's disease, it is uncomfortable to put it alongside ADAS-cog. Refer to the FDA guidance for industry published in 2018.

Reviewer #2: The authors carefully review the currently available evidence regarding amyloid-targeted pharmacotherapy for early AD, based on awareness of academic/social controversy about unfruitful results of the previous AD pharmacotherapy clinical trials. The reviewer raises some points to be clarified, possibly to improve the manuscript for better-understanding to the readers：

Types of outcome measures

“We are not interested in changes in biomarkers (Cerebrospinal fluid (CSF) indications of beta-amyloid 42, Tau, and

phospho-tau) therefore, changes in biomarkers are not included in our study.”

- In focusing pharmacotherapeutic potentials of AD trials, these important biological information is essential to be reviewed. Authors should include these biomarkers to the outcome measures in meta-analysis.

Study selection

“To facilitate the screening process, the team will use Covidence (see www.covidence.org), a web-based tool designed to support systematic review screening and manage references and full-text PDFs. Duplicates will be removed by the Covidence software.”

- Authors should explain more clearly what type of process will be handled by Covidence staff.

Data extraction

“Any discrepancies between the two reviewers were resolved by consensus and by consulting a third reviewer.”

- Authors should describe clearly who will be the candidate of the third reviewer.

Quantitative synthesis

“We will calculate the I2 to test for heterogeneity for each pooled result. We will pool our results based on each outcome of interest (ADAS-Cog 11, ADAS-Cog 12, ADAS-Cog 13, ADASCog 14, CDR-SB, MMSE, EQ-5D, EQ-5D (Proxy), QoL-AD, ADCS-ADL, and NPI).”

- It is known that there are some discrepancies of the scores in the versions of NPI, NPI-Q and NPI-NH. Authors should clarify what version of NPI was used in each clinical trial.

7. PLOS authors have the option to publish the peer review history of their article (what does this mean?). If published, this will include your full peer review and any attached files.

Reviewer #1: No

Reviewer #2: No

---

## [Author Response · Author response to Decision Letter 0]

3 Feb 2024

Dear Editor-in-Chief,

I hope this letter finds you well. We appreciate the thorough review of our manuscript titled "Do pharmacological interventions targeting beta-amyloid improve cognition, physical functioning, and overt behaviour of Alzheimer’s Disease (AD) patients: A protocol for a meta-analysis of Phase 3 clinical trials both completed and terminated," and we thank the reviewers for their insightful comments and suggestions. We have carefully considered each of the reviewer's points and offer our detailed responses below:

Reviewer: This manuscript addresses a protocol for a forthcoming meta-analysis of anti-Aβ therapy for Alzheimer's disease. While the results of the meta-analysis are important, it is difficult to judge the significance of publishing the protocol because the method itself of the meta-analysis is not novel and only items are modified.

Response: We acknowledge the reviewer's concern regarding the novelty of the methodology. However, we believe that the new clinical trial data available since the latest systematic reviews of anti-Aβ therapy for Alzheimer's disease warrants an updated review. Furthermore, our systematic review is distinctive in its inclusion of all three domains (cognitive outcomes, outcomes of physical functioning and behavior) and all stages of AD, which to our knowledge, has not been previously attempted. This broader approach will provide a more comprehensive understanding of the effectiveness of anti-Aβ therapy across diverse aspects of Alzheimer's disease.

Reviewer: Although grouped together as anti-Aβ drugs in the protocol, the mechanisms of drugs that inhibit Aβ production and those that promote Aβ clearance are very different, and I am concerned that the scientific significance of the results may be diminished by grouping them together as "anti-Aβ drugs".

Response: We appreciate the reviewer's concern regarding the grouping of different mechanisms of anti-Aβ drugs. We have taken into account the distinction between drugs that inhibit Aβ production and those that promote Aβ clearance. Our systematic review will categorize and analyze data separately for each intervention type to address this concern. The purpose of including both intervention types is to comprehensively explore the Aβ hypothesis and provide insights into their distinct effects on Alzheimer's disease.

Reviewer: The preclinical stage is just the clinically asymptomatic stage, which can include A+T+N- and even A+T+N+ as well as A+T-N-.

Response: We acknowledge the reviewer's point and have removed the reference to the preclinical stage from the text.

Reviewer: "BETA1" is a typo for BACE1 (beta-site amyloid precursor protein cleaving enzyme 1).

Response: We apologize for the typographical error and have corrected it to accurately reference BACE1 in the manuscript.

Reviewer: Although many have criticized the approval of Aduhelm because of the ambiguity of the decision criteria by the FDA, I guess no one has described it as "it is a step backwards".

Response: We understand the reviewer's perspective and have removed the phrase "it is a step backwards" to ensure accuracy and clarity in our manuscript.

Reviewer: At the time of submission, Lecanemab was already approved in addition to Aducanumab, so that should also be listed.

Response: We appreciate the reviewer's input and will conduct an updated search to include Lecanemab in our analysis, as per the protocol.

Reviewer: Since phase III trials of anti-Aβ drugs were not initiated until 2007 or later, it is more likely that the 2011 NIA-AA criteria are used in the trials rather than the older NINCDS-ADRDA criteria.

Response: We acknowledge the reviewer's comment and have reviewed our sources more carefully. While the majority of trials may indeed use the 2011 NIA-AA criteria, we have found instances where the older NINCDS-ADRDA criteria were used. We will provide further clarification in the manuscript regarding the criteria used in the trials.

Reviewer: CDR-SB is a composite of cognitive function and global function (probably equal to "physical function" in the article), so for those familiar with clinical trials in Alzheimer's disease, it is uncomfortable to put it alongside ADAS-cog. Refer to the FDA guidance for industry published in 2018.

Response: We appreciate the reviewer's suggestion and understand the concern. Our intention is to provide a comprehensive overview of outcome measures used in clinical trials. While we acknowledge the difference between CDR-SB and ADAS-cog, we will retain the information to offer a holistic view of the measures employed in Alzheimer's disease trials. This will also allow readers to see which measures are sensitive to the drugs targeting Aβ.

Reviewer: Types of outcome measures: “We are not interested in changes in biomarkers (Cerebrospinal fluid (CSF) indications of beta-amyloid 42, Tau, and phospho-tau) therefore, changes in biomarkers are not included in our study.” In focusing pharmacotherapeutic potentials of AD trials, these important biological information is essential to be reviewed. Authors should include these biomarkers in the outcome measures in the meta-analysis.

Response: We acknowledge the importance of biomarker data in Alzheimer's disease trials. Considering this, we have revised our approach and will include changes in biomarkers, specifically CSF indications of beta-amyloid 42, Tau, and phospho-tau, in our outcome measures for the meta-analysis.

Reviewer: Study selection: “To facilitate the screening process, the team will use Covidence (see www.covidence.org), a web-based tool designed to support systematic review screening and manage references and full-text PDFs. Duplicates will be removed by the Covidence software.” Authors should explain more clearly what type of process will be handled by Covidence staff.

Response: We apologize for any confusion and appreciate the reviewer's feedback. To clarify, the screening process will be solely conducted by the authors using the Covidence software. No external staff or assistance will be involved in this process.

Reviewer: Data extraction: “Any discrepancies between the two reviewers were resolved by consensus and by consulting a third reviewer.” Authors should describe clearly who will be the candidate of the third reviewer.

Response: We appreciate the reviewer's suggestion and will explicitly mention the initials of the third reviewer who will be consulted in case of discrepancies during data extraction.

Reviewer: Quantitative synthesis: “We will calculate the I2 to test for heterogeneity for each pooled result. We will pool our results based on each outcome of interest (ADAS-Cog 11, ADAS-Cog 12, ADAS-Cog 13, ADAS-Cog 14, CDR-SB, MMSE, EQ-5D, EQ-5D (Proxy), QoL-AD, ADCS-ADL, and NPI).” It is known that there are some discrepancies of the scores in the versions of NPI, NPI-Q and NPI-NH. Authors should clarify what version of NPI was used in each clinical trial.

Response: We acknowledge the reviewer's concern regarding the discrepancies in versions of NPI. In our manuscript, we will clearly specify the version of NPI used in each clinical trial to ensure transparency and accuracy in our analysis.

We thank the reviewers for their valuable feedback, which has undoubtedly improved the quality and rigor of our manuscript. We have carefully addressed each concern and made the necessary revisions to ensure the accuracy, clarity, and scientific merit of our work. We hope these clarifications and amendments

Thank you,

Chelsea

---

## [Decision Letter · Decision Letter 1]

16 Apr 2024

PONE-D-23-05935R1Do beta-amyloid-targeted interventions improve cognition, physical functioning, and overt behaviour of Alzheimer’s Disease (AD) patients: A protocol for a meta-analysis of Phase 3 clinical trials both completed and terminatedPLOS ONE

Dear Dr. Stellick (Bedrejo),

Thank you for submitting your manuscript to PLOS ONE. After careful consideration, we feel that it has merit but does not fully meet PLOS ONE’s publication criteria as it currently stands. Therefore, we invite you to submit a revised version of the manuscript that addresses the points raised during the review process.

We look forward to receiving your revised manuscript.

Kind regards,

Kensaku Kasuga

Academic Editor

PLOS ONE

Journal Requirements:

Reviewers' comments:

Reviewer's Responses to Questions

**Comments to the Author**

1. Does the manuscript provide a valid rationale for the proposed study, with clearly identified and justified research questions?

Reviewer #1: Yes

Reviewer #2: Yes

2. Is the protocol technically sound and planned in a manner that will lead to a meaningful outcome and allow testing the stated hypotheses?

Reviewer #1: Yes

Reviewer #2: Yes

3. Is the methodology feasible and described in sufficient detail to allow the work to be replicable?

Reviewer #1: Yes

Reviewer #2: Yes

4. Have the authors described where all data underlying the findings will be made available when the study is complete?

Reviewer #1: Yes

Reviewer #2: Yes

5. Is the manuscript presented in an intelligible fashion and written in standard English?

Reviewer #1: Yes

Reviewer #2: Yes

6. Review Comments to the Author

You may also provide optional suggestions and comments to authors that they might find helpful in planning their study.

Reviewer #1: The author partially complied with the reviewers' comments and revised the manuscript. Let me point out only a few minor points.

1) Please include "Eisai's lecanemab" in P.4, L.9. Since lecanemab is the only drug that is fully approved and already widely used in clinical practice, it is unnatural that it is not mentioned here.

2) As another reviewer pointed out, please add "2011 NIA-AA criteria" alongside the NINCDS-ADRDA criteria to the last paragraph of P.6 and inclusion criteria in Table 2. As the author mentioned in response comments, older trials (the number should be larger) used the NINCDS-ADRDA criteria. However, the NINCDS-ADRDA criteria is getting less familiar to inexperienced researchers. This is just to make the paper and its value understandable to more researchers.

3) If it is appropriate, please mention the period during which the paper to be extracted was published. (ex. "published from xx to xx", "published by 2024 xx")

Reviewer #2: The authors responded to all the points raised by the reviewer to be clarified. The reviewer recognizes that the qualitiy of the revised manuscript has enough to be considered to be published in PLoSone.

7. PLOS authors have the option to publish the peer review history of their article (what does this mean?). If published, this will include your full peer review and any attached files.

Reviewer #1: No

Reviewer #2: No

---

## [Author Response · Author response to Decision Letter 1]

30 Jun 2024

The suggestions from the review have been appropriately incorporated into the new manuscript.

---

## [Editor Report · Decision Letter 2]

11 Jul 2024

Do beta-amyloid-targeted interventions improve cognition, physical functioning, and overt behaviour of Alzheimer’s Disease (AD) patients: A protocol for a meta-analysis of Phase 3 clinical trials both completed and terminated

PONE-D-23-05935R2

Dear Dr. Stellick (Bedrejo),

We’re pleased to inform you that your manuscript has been judged scientifically suitable for publication and will be formally accepted for publication once it meets all outstanding technical requirements.

Kind regards,

Kensaku Kasuga

Academic Editor

PLOS ONE
---

## [Editor Report · Acceptance letter]

9 Aug 2024

PONE-D-23-05935R2 

PLOS ONE

Dear Dr. Stellick (Bedrejo), 

I'm pleased to inform you that your manuscript has been deemed suitable for publication in PLOS ONE. Congratulations! Your manuscript is now being handed over to our production team.

Kind regards, 

on behalf of

Dr. Kensaku Kasuga 

Academic Editor

PLOS ONE